# No tension between assembly models of super massive black hole binaries and pulsar observations

Hannah Middleton[1], Siyuan Chen[1], Walter Del Pozzo[1,2], Alberto Sesana[1] & Alberto Vecchio[1]

Pulsar timing arrays are presently the only means to search for the gravitational wave stochastic background from super massive black hole binary populations, considered to be within the grasp of current or near-future observations. The stringent upper limit from the Parkes Pulsar Timing Array has been interpreted as excluding (>90% confidence) the current paradigm of binary assembly through galaxy mergers and hardening via stellar interaction, suggesting evolution is accelerated or stalled. Using Bayesian hierarchical modelling we consider implications of this upper limit for a range of astrophysical scenarios, without invoking stalling, nor more exotic physical processes. All scenarios are fully consistent with the upper limit, but (weak) bounds on population parameters can be inferred. Recent upward revisions of the black hole–galaxy bulge mass relation are disfavoured at $1.6\sigma$ against lighter models. Once sensitivity improves by an order of magnitude, a non-detection will disfavour the most optimistic scenarios at $3.9\sigma$.

[1] Institute of Gravitational Wave Astronomy and School of Physics and Astronomy, University of Birmingham, Birmingham B15 2TT, UK. [2] Dipartimento di Fisica "Enrico Fermi", Università di Pisa and INFN sezione di Pisa, Pisa I-56127, Italy. Correspondence and requests for materials should be addressed to H.M. (email: hannahm@star.sr.bham.ac.uk)

Dedicated timing campaigns of ultra-stable radio pulsars lasting over a decade and carried out with the best radio telescopes around the globe have targeted the isotropic gravitational-wave (GW) background in the frequency region $\sim 10^{-9}$–$10^{-7}$ Hz generated by the cosmic population of merging super massive black hole binaries (SMBHBs). In the hierarchical clustering scenario of galaxy formation, galaxies form through a sequence of mergers[1]. In this process, the SMBHs hosted at their centers will inevitably form a large number of binaries[2], forming an abundant population of GW sources in the Universe. Detecting and/or placing constraints on their emitted signal will thus provide an insight into the formation and evolution of SMBHs in connection with their galaxy hosts and will help to better understand the role played by SMBHs in galaxy evolution and the dynamical processes operating during galaxy mergers (for a review see ref. [3]).

No detection at nHz frequencies has been reported so far. The most stringent constraint on an isotropic background radiation has been obtained through an 11-year-long timing of 4 radio-pulsars by the Parkes Pulsar Timing Array (PPTA). It yields an upper limit on the GW characteristic amplitude of $h_{1yr} = 1.0 \times 10^{-15}$ (at 95% confidence) at a frequency of 1 yr[14]. Consistent results, although a factor $\approx 2$ less stringent, have also been reported by the European PTA[5], the North American Nanohertz Observatory for Gravitational Waves[6] and the International PTA[7], an international consortium of the three regional PTA collaborations. Those values are in the range of signal amplitudes predicted by state-of-the-art SMBHB population models, and can therefore be used to constrain such a population. It has been noted, however, that these limits start to be sensitive to uncertainties in the determination of the solar system ephemeris used in the analysis. Recent unpublished work has in fact found that different ephemeris choices can result in a partial degradation of the upper limit[8]. This is still an active area of research which may lead to a small upward revision of the upper limit, a circumstance which, if anything, will strengthen the conclusion of our analysis. Here we consider the most stringent upper limit from the PPTA in order to glean what can be learnt at this stage and also determine whether current SMBHB population models are indeed cast into doubt.

Using the PPTA limit, we place bounds on the properties of the sub-parsec population of cosmic SMBHBs (in the mass range $\sim 10^7$–$10^{10}$ M$_\odot$) and explore what constraints, if any, can be put on the salient physical processes that lead to the formation and evolution of these objects. We consider a comprehensive suite of astrophysical models that combine observational constraints on the SMBHB population with state-of-the-art dynamical modelling of binary evolution. The SMBHB merger rate is anchored to observational estimates of the host galaxy merger rate by a set of SMBH–host relations (see refs. [9,10] and Methods). Rates obtained in this way are well captured by a five parameter analytical function of mass and redshift, once model parameters are restricted to the appropriate prior range (see Methods). Individual binaries are assumed to hold a constant eccentricity so long as they evolve via three-body scattering and gradually circularize once GW emission takes over. Their dynamical evolution and emission properties are regulated by the density of the stellar environment (assumed to be a Hernquist profile[11] with total mass determined by the SMBH mass–galaxy bulge mass relation) and by the eccentricity during the three-body scattering phase, which we take as a free parameter. For each set of model parameters, the characteristic GW strain $h_c(f)$ at the observed frequency $f$ is computed as described in ref. [12] and summarized in Methods. Our model encapsulates the significant uncertainties in the GW background due to the poorly constrained SMBHB merger rate and has the flexibility to produce a low frequency turnover due to

either three-body scattering or high eccentricities. SMBHBs are assumed to merge with no significant delay after galaxies merge. As such, the models do not include the effect of stalling or delayed mergers[13].

We find that although PTAs have well and truly achieved a sensitivity for which detection is possible based on model predictions, the present lack of a detection provides no reason to question these models. We highlight the impact of the SMBH-galaxy relation by considering a selection of models which cover the entire range of the predicted background amplitude. To be definitive, we consider: (i) an optimistic model (here labelled KH13, based on ref. [14]), which provides a prediction of the GW background with median amplitude at $f = 1$ yr$^{-1}$ of $h_{1yr} = 1.5 \times 10^{-15}$; (ii) a conservative model (labelled G09, based on ref. [15]), with median $h_{1yr} = 7 \times 10^{-16}$; (iii) an ultra-conservative model (labelled S16, based on ref. [16]), with $h_{1yr} = 4 \times 10^{-16}$; and finally (iv) a model that spans the whole range of predictions within our assumptions (which we label 'ALL'). It is noteworthy that the latter contains as subsets KH13, G09 and S16, but it is not limited to them. Moreover, model 'ALL' spans an $h_{1yr}$ amplitude range that comfortably include GW backgrounds estimated by other authors employing different techniques (e.g, see refs. [17–20]). Details on the models are provided in Methods. We find all models to be consistent with the current PTA upper limits.

## Results

**Inference using the upper limit**. For each model, we use a Bayesian hierarchical analysis to compute the model evidence (which is the probability of the model given the data and allows for the direct comparison of models) and posterior density functions on the model parameters given the observational results reported by ref. [4]. We find that the upper limit is now beginning to probe the most optimistic predictions, but all models are so far consistent with the data. Figure 1, our main result, compares the predictions under different model assumptions with the observed upper limit. The dotted area shows the prior range of the GW amplitude under the model assumptions, and the orange solid line shows the 95% confidence PPTA upper limit on $h_c$. The (central) 68% and 90% posterior probability intervals on $h_c$ are shown by the shaded blue bands. The posterior density functions (PDFs) on the right hand side of each plot gives the prior (black dashed line) and posterior (blue line) for $h_c$ at a reference frequency of $f \sim 1/5$ yr$^{-1}$.

The difference between the dotted region and the shaded bands in the main panels in Fig. 1 indicates the constraining power of the Parkes PTA limit on astrophysical models—the greater the difference between the two regions, the smaller is the consistency of that particular model with the data. We see that although some upper portion of the allowable prior region is removed from the 90% posterior probability interval (less so for S16), none of the models can be ruled out at any significant level. The confidence bands across the frequency range are constructed by taking the relevant credibility region of the posterior distribution of $h_c$ at each frequency, and therefore the boundaries of each band do not follow any particular functional form as a function of frequency. In addition, although eccentricity is allowed by the data, the power-law spectrum of circular binaries driven by radiation reaction alone can clearly be consistently placed within these bands (see also Supplementary Fig. 1 for further details on the individual parameter posteriors including eccentricity). This can be quantified in terms of the model evidences $\mathcal{Z}$, shown in Table 1. The normalization is chosen so that a putative model unaffected by the limit yields $\mathcal{Z} = 1$ and therefore the values can be interpreted as Bayes factors against such a model. None of the posterior probabilities of the models with respect to this putative one show any tension. As an example,

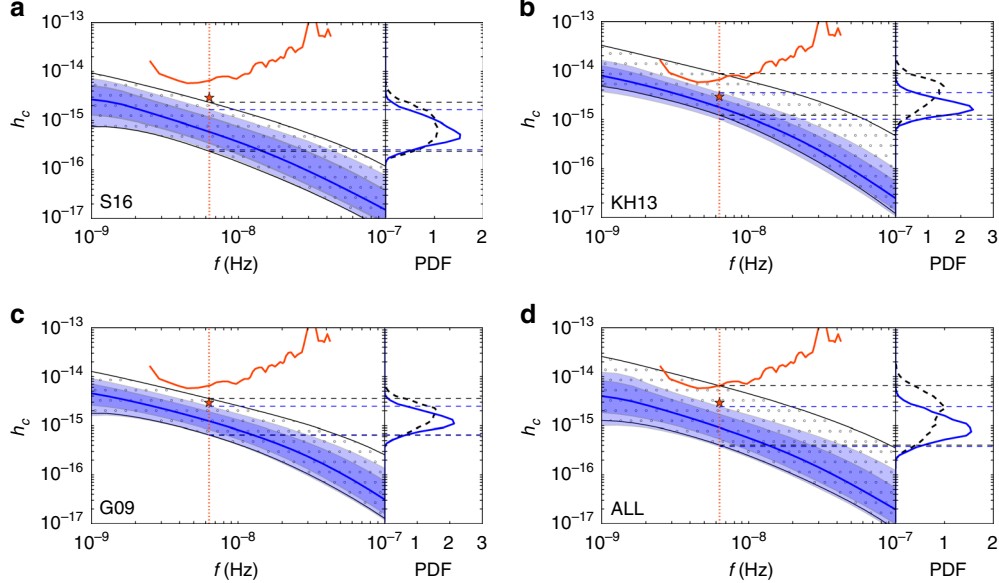

**Fig. 1** The posterior density function on the gravitational wave characteristic amplitude. The four panels compare the prior and posterior density functions on the GW stochastic background characteristic amplitude in light of the PPTA upper limit for each of the astrophysical models considered here: **a** S16; **b** KH13; **c** G09; **d** ALL. The central 90% region of the prior is indicated by the black dotted band and the posterior is shown by the progressively lighter blue shading indicating the central 68% and 90% regions, respectively, along with the median (solid blue line). Also shown are the PPTA bin-by-bin limit (orange solid line) and the corresponding integrated limit assuming $h_c(f) \propto f^{-2/3}$ (orange star and vertical dotted line). The difference in the prior and posterior indicates how much has been learnt from the PPTA data. In each panel, the right-hand side one-dimensional distribution shows the prior (black dashed) and posterior (blue solid) at a reference frequency of $f \sim 1/5 \, \mathrm{yr}^{-1}$, with the central 90% regions marked (black and blue dashed lines respectively)

### Table 1 Kullback–Leibler divergences and evidences for different models

| Model | $h_{1\mathrm{yr}} = 1 \times 10^{-15}$ (PPTA) | | $h_{1\mathrm{yr}} = 3 \times 10^{-16}$ | | $h_{1\mathrm{yr}} = 1 \times 10^{-16}$ | |
|---|---|---|---|---|---|---|
| | K-L divergence | $\log \mathcal{Z}$ | K-L divergence | $\log \mathcal{Z}$ | K-L divergence | $\log \mathcal{Z}$ |
| KH13 | 0.85 | −2.36 | 2.25 | −5.68 | 5.18 | −13.17 |
| G09 | 0.39 | −1.2 | 1.11 | −3.35 | 2.86 | −8.26 |
| S16 | 0.37 | −0.6 | 0.69 | −1.62 | 1.42 | −3.82 |
| ALL | 0.62 | −1.23 | 1.33 | −2.68 | 2.50 | −5.74 |

The values in the table show the K-L divergence and natural logarithm of the evidence, $\log \mathcal{Z}$, for each of the four astrophysical models given the PPTA upper limit at $h_{1\mathrm{yr}} = 1 \times 10^{-15}$ and for more stringent putative limits at the levels of $3 \times 10^{-16}$ and $1 \times 10^{-16}$

for models ALL and S16 we find $e^{-1.23} = 0.3$ and $e^{-0.6} = 0.55$, respectively. Similar conclusions can be drawn from the Kullback–Leibler (K-L) divergences between the prior and posterior on the characteristic amplitude for a given model (with which we measure the difference between the prior and posterior). For models ALL and S16, these yield 0.62 and 0.37, respectively. As a comparison, these values correspond to the K-L divergence between two Gaussian distributions with the same variance and means approximately 1.1 (for ALL) and 0.8 (for S16) SD apart (the K-L divergence between two normal distributions $p \sim N\left(\mu_p, \sigma_p^2\right)$ and $q \sim N\left(\mu_q, \sigma_q^2\right)$ is $D_{\mathrm{KL}}(p||q) = \ln(\sigma_q/\sigma_p) - 1/2 + 1/2\left[(\sigma_p/\sigma_q)^2 + \left(\mu_p - \mu_q\right)^2/\sigma_q^2\right]$. For $\sigma_p = \sigma_q$ and $\mu_p = \mu_q + \sigma_q$ the K-L divergence is 0.5).

Figure 2 summarizes the natural logarithm of the ratio of the model evidences, i.e. the Bayes factors, between all possible combinations of models and also the K-L divergences whose numerical values are listed in Table 1. Both metrics clearly indicate that there is little to choose from between the models. The least favoured model in the range of those considered here is KH13, with Bayes factors in favour of the others ranging from ≈

1.13 to ≈ 1.76. These are however values of order unity and no decisive inference can be made from the data[21]. Comparisons between each of the individual model parameters (see Methods) posterior and prior distribution functions are described in Supplementary Fig. 1 and Supplementary Table 1, which further support our conclusions. For KH13, the model that produces the strongest GW background, we find a probability of $e^{-2.36} = 0.094$ with respect to a putative model that is unaffected by the limit. KH13 is therefore disfavoured at ~1.6σ. This conclusion is reflected in the value of the K-L divergence of 0.85 (this is the same K-L divergence as between two Gaussian distributions with the same variance and means ~1.3 standard deviation apart). We note that ref. [4] choose in their analysis only a sub-sample of the ref. [9] models, with properties similar to KH13. Our results for KH13 are therefore consistent with the 91%-to-97% 'exclusion' claimed by ref. [4].

### Discussion

It is argued in ref. [4] that the Parkes PTA upper-limit excludes at high confidence standard models of SMBH assembly—i.e, those considered in this work—and therefore these models need to be substantially revised to accommodate either accelerated mergers

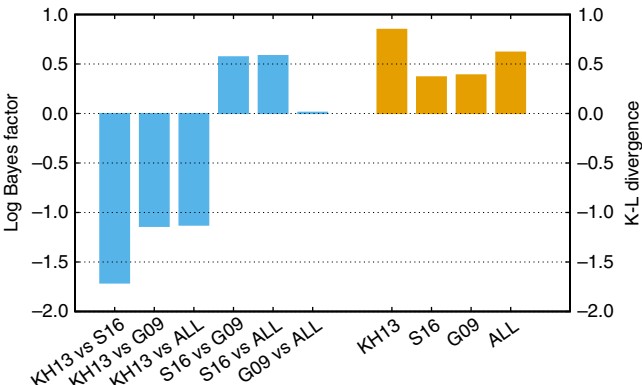

**Fig. 2** Bayes factors and Kullback–Leibler divergences for different models. We compare the Bayes factors between model pairs (left hand, blue bars) and the Kullback–Leibler (K-L) divergences between the prior and posterior of the characteristic amplitude (right hand, orange bars). The small range of Bayes factors indicates that there is little to choose from between these models, although KH13 is weakly disfavoured against the others. The K-L divergences also support this conclusion. Although all values are small, KH13 has the largest K-L divergence (greatest difference between prior and posterior) of the four models

via strong interaction with the environment or inefficient SMBHB formation following galaxy mergers. The work presented here does not support either claim. In particular, the posterior parameter distributions (see Supplementary Fig. 1) favour neither high eccentricities nor particularly high stellar densities, indicating that a low frequency spectral turnover induced by SMBHB dynamics is not required to reconcile the PTA upper limit with existing models. Similar to ref. 22, this finding does not support an observing strategy revision in favour of higher cadence observations aimed at improving the high frequency sensitivity, as proposed by ref. 4. Likewise, neither stalling nor delays between galaxy and SMBHB mergers, which, by construction, are not included in the models considered here, are needed to explain the lack of a detection of GWs at the present sensitivity level. Compared with previous analyses, our work implies a stronger rejection of the statement that there is tension between PTA data and theoretical SMBHB population models. For example, ref. 13 invoked time delays to reconcile the PPTA upper limit with selected SMBH-galaxy relations, however they assume a narrow range of possible SMBHB merger histories and do not consider SMBHB dynamics. The analysis of ref. 6 tends to favour a spectral turnover due to either high eccentricity or strong environmental coupling, however they use a simplified analysis where each relevant physical parameter is accounted for separately. When allowing all the parameters to vary simultaneously, we find that none of them has a critical impact on the inference, and current SMBHB population models are broadly consistent with the PTA upper limits, without the need to invoke a low-frequency spectral turnover.

On the other hand, PTA limits are now starting to provide interesting information about the population of merging SMBHs. The fact that KH13 is disfavoured at $1.4\sigma$ with respect to S16 indicates that the population may have fewer high mass binaries, mildly favouring SMBH-host galaxy relations with lower normalizations. This indicates that the gravitational wave background level is likely below the $10^{-15}$ level, making detection difficult with current telescopes. In this respect, our analysis highlights the importance of upcoming facilities such as Meer-KAT[23], FAST[24] and the Square Kilometer Array (SKA)[25]. Their superior timing capabilities, together with their survey potential

in finding new stable millisecond pulsars, will provide the necessary ground to improve sensitivity down to $h_{1yr} \sim 10^{-16}$, which is in line with the lower limit of the expected stochastic gravitational wave background according to our current understanding of SMBH evolution[26]. Although not yet decisive, our findings highlight the potential of PTAs in informing the current debate on the SMBH-host galaxy relation. Recent discoveries of over-massive black holes in brightest cluster ellipticals[27,28] led to an upward revision of those relations[14,29]. However, several authors attribute the high normalization of the recent SMBH-host galaxy relations to selection biases[16] or to the intrinsic difficulty of resolving the SMBH fingerprint in measurements based on stellar dynamics (see discussion in ref. 30). Future facilities such as the Extremely Large Telescope[31] and the Thirty Meter Tele-scope[32] will likely measure many more SMBH masses in elliptical galaxies[33], providing a better understanding of the SMBH-host galaxy relations. PTA limits may therefore be used to gain more information about the other underlying uncertainties in the model, in particular the massive galaxy merger rate, which is currently poorly constrained observationally (e.g, see refs 34,35).

An important question is: what is the sensitivity level required to really put under stress our current understanding of SMBHB assembly? If a null result persists in PTA experiments, this will in turn lead to a legitimate re-thinking of the PTA observing strategy to target possibly more promising frequencies of the GW spectrum. To address this question, we simulate future sensitivity improvements by shifting the Parkes PTA sensitivity curve down to provide 95% upper limits of $h_{1yr}$ at $3 \times 10^{-16}$ and $1 \times 10^{-16}$. The results are summarized in Table 1 and more details are provided in Supplementary Fig. 2, Supplementary Table 2 and Supplementary Note 1. At $3 \times 10^{-16}$, possibly within the sensitivity reach of PTAs in the next $\approx 5$ years, S16 will be significantly favoured against KH13, with a Bayes factor of $e^{4.06}$, and only marginally favoured over G09, with Bayes factor of $e^{1.76}$. It will still be impossible to reject this model at any reasonable significant level with respect to, say, a model which predicts negligible GW background radiation at $\sim 10^{-9}-10^{-8}$ Hz. However SMBH–host galaxy relations with high normalizations will show a $\approx 2\sigma$ tension with more conservative models. At $1 \times 10^{-16}$, within reach in the next decade with the advent of MeerKAT, FAST and SKA, models KH13, G09 and ALL are disfavoured at $3.9\sigma$, $2.5\sigma$ and $1.2\sigma$, respectively, in comparison with S16. K-L divergences in the range 5.18–1.42 show that the data are truly informative. S16 is also disfavoured at $2.3\sigma$ with respect to a model unaffected by the data, possibly indicating the need of additional physical processes to be included in the models.

## Methods

**Analytical description of the GW background.** The GW background from a cosmic population of SMBHBs is determined by the binary merger rate and by the dynamical properties of the systems during their inspiral. The comoving number density of SMBHBs per unit log chirp mass ($\mathcal{M} = (M_1 M_2)^{3/5}/(M_1 + M_2)^{1/5}$) and unit redshift, $d^2n/(d \log_{10} \mathcal{M} dz)$, defines the normalization of the GW spectrum. If all binaries evolve under the influence of GW backreaction only in circular orbits, then the spectral index is fixed at $h_c(f) \propto f^{-2/3}$ and the GW background is fully determined[36]. However, to get to the point at which GW emission is efficient, SMBHBs need to exchange energy and angular momentum with their stellar and/or gaseous environment[3], a process that can lead to an increase in the binary eccentricity (e.g., see refs 37,38.). We assume SMBHBs evolve via three-body scattering against the dense stellar background up to a transition frequency $f_t$ at which GW emission takes over. According to recent studies[39,40], the hardening is dictated by the density of background stars $\rho_i$ at the influence radius of the binary $r_i$. The bulge stellar density is assumed to follow a Hernquist density profile[11] with total mass $M_*$ and scale radius $a$ determined by the SMBHB total mass $M = M_1 + M_2$ via empirical relations from the literature (see full details in ref. 12). Therefore, for each individual system, $\rho_i$ is determined solely by $M$. In the stellar hardening phase, the binary is assumed to hold constant eccentricity $e_t$ up to $f_t$, beyond which it circularizes under the effect of the now dominant GW backreaction. The GW spectrum emitted by an individual binary adiabatically inspiralling under these assumptions behaves as $h_c(f) \propto f$ for $f \ll f_t$ and settles to the standard $h_c(f) \propto f^{-2/3}$

for $f \gg f_t$. The spectrum has a turnover around $f_t$ and its exact location depends on the binary eccentricity $e_t$. The observed GW spectrum is therefore uniquely determined by the binary chirp mass $\mathcal{M}$, redshift $z$, transition frequency $f_t$ and eccentricity at transition $e_t$.

The GW spectrum from the overall population can be computed by integrating the spectrum of each individual system over the co-moving number density of merging SMBHBs

$$h_c^2(f) = \int dz \int d\log_{10}\mathcal{M} \frac{d^2 n}{d\log_{10}\mathcal{M} dz}$$
$$\times h_{c,fit}^2 \left( f \frac{f_{p,0}}{f_{p,t}} \right) \left( \frac{f_{p,t}}{f_{p,0}} \right)^{-4/3} \left( \frac{\mathcal{M}}{\mathcal{M}_0} \right)^{5/3} \left( \frac{1+z}{1+z_0} \right)^{-1/3} \quad (1)$$

where $h_{c,fit}$ is an analytic fit to the GW spectrum of a reference binary with chirp mass $\mathcal{M}_0$ at redshift $z_0$ (i.e., assuming $d^2 n/(d\log_{10}\mathcal{M} dz) = \delta(\mathcal{M} - \mathcal{M}_0)\delta(z - z_0)$), characterized by an eccentricity of $e_0$ at a reference frequency $f_0$. For these reference values, the peak frequency of the spectrum $f_{p,0}$ is computed. The contribution of a SMBHB with generic chirp mass, emission redshift, transition frequency $f_t$ and initial eccentricity $e_t$ are then simply computed by calculating the spectrum at a rescaled frequency $f(f_{p,0}/f_{p,t})$ and by shifting it with frequency mass and redshift as indicated in Eq. (1). In [12] it was demonstrated that this simple self-similar computation of the GW spectrum is sufficient to describe the expected GW signal from a population of eccentric SMBHBs driven by three-body scattering at $f > 1$ nHz, relevant to PTA measurement.

As stated above, the shape of the spectrum depends on $\rho_i$ and $e_t$. The stellar density $\rho_i$ regulates the location of $f_t$; the denser the environment, the higher the transition frequency. SMBHBs evolving in extremely dense environments will therefore show a turnover in the GW spectrum at higher frequency. The effect of $e_t$ is twofold. On the one hand, eccentric binaries emit GWs more efficiently at a given orbital frequency, thus decoupling at lower $f_t$ with respect to circular ones. On the other hand, eccentricity redistributes the emitted GW power at higher frequencies, thus pushing the spectral turnover to high frequencies. In our default model, $\rho_i$ is fixed by the SMBHB total mass $M$ and we make the simplifying assumption that all systems have the same $e_t$. We also consider an extended model where $\rho_i$ is multiplied by a free parameter $\eta$. This corresponds to a simple rescaling of the central stellar density, relaxing the strict $M - \rho_i$ relation imposed by our default model. We stress here that including this parameter in our main analysis yielded quantitatively identical results.

We use a generic simple model for the cosmic merger rate density of SMBHBs based on an overall amplitude and two power law distributions with exponential cutoffs,

$$\frac{d^2 n}{d\log_{10}\mathcal{M} dz} = \dot{n}_0 \left( \frac{\mathcal{M}}{10^7 M_\odot} \right)^{-\alpha} \exp\left( -\frac{\mathcal{M}}{\mathcal{M}_*} \right) (1+z)^\beta \exp\left( -\frac{z}{z_*} \right) \frac{dt_r}{dz} \quad (2)$$

where $dt_r/dz$ is the relationship between time and redshift assuming a standard $\Lambda$CDM flat Universe with cosmological constant of $H_0 = 70$ km s$^{-1}$ Mpc$^{-1}$. The five free parameters are: $\dot{n}_0$ representing the co-moving number of mergers per Mpc$^3$ per Gyr; $\alpha$ and $\mathcal{M}_*$ control the slope and cutoff of the chirp mass distribution respectively; $\beta$ and $z_*$ regulate the equivalent properties of the redshift distribution. Eq (2) is also used to compute the number of emitting systems per frequency resolution bin at $f > 10$ nHz. The small number statistics of the most massive binaries determines a steepening of the GW spectrum at high frequencies, full details of the computation are found in refs. [41] and [12]. The GW spectrum is therefore uniquely computed by a set of six(seven) parameters $\theta = \dot{n}_0, \beta, z_*, \alpha, \mathcal{M}_*, e_t(,\eta)$.

**Anchoring the model before astrophysical models**. Although no sub-parsec SMBHBs emitting in the PTA frequency range have been unambiguously identified to date, their cosmic merger rate can be connected to the merger rate of their host galaxies. The procedure has been extensively described in ref. [9]. The galaxy merger rate can be estimated directly from observations via

$$\frac{d^3 n_G}{dz dM_G dq} = \frac{\phi(M_G, z)}{M_G \ln 10} \frac{F(z, M_G, q)}{\tau(z, M_G, q)} \frac{dt_r}{dz}. \quad (3)$$

Here, $M_G$ is the galaxy mass; $\phi(M_G, z) = (dn/d\log M_G)_z$ is the galaxy mass function measured at redshift $z$; $F(M_G, q, z) = (df_p/dq)_{M_G,z}$, for every $M_G$ and $z$, denotes the fraction of galaxies paired with a companion galaxy with mass ratio between $q$ and $q + \delta q$; $\tau(z, M_G, q)$ is the merger timescale of the pair as a function of the relevant parameters. We construct a library of galaxy merger rates by combining four measurements of the galaxy mass function $\phi(M_G, z)$[42–45], four estimates of the close pair fraction $F(M_G, q, z)$[46–49] and two estimates of the merger timescale $\tau(z, M_G, q)$[50,51]. For each of the galaxy mass functions and pair fractions, we consider three estimates given by the best fit and the two boundaries of the $1\sigma$ confidence interval reported by the authors. We therefore have $12 \times 12 \times 2 = 288$ galaxy merger rates. Each merging galaxy pair is assigned SMBHs with masses drawn from 14 different SMBH–galaxy relations found in the literature, for more details see Supplementary Table 3. SMBHBs are assumed to merge in coincidence with the host galaxies (i.e., no stalling or extra delays), but can accrete either before or after merger according to the three different prescriptions described in ref. [52]. This gives a total of $14 \times 3 = 42$ distinctive SMBH populations for a given galaxy merger model. We combine the 288 galaxy merger rates as per Eq. (3) and the 42 SMBH masses assigned via using Supplementary Table 3, plus accretion prescriptions into a grand total of 12,096 SMBHB population models. Given the uncertainties, biases, selection effects, and poor understanding on the underlying physics affecting each of the individual ingredients, we do not attempt a ranking of the models, and give each of them equal weight. The models result in an allowed SMBHB merger rate density as a function of chirp mass and redshift.

We then marginalize over mass and redshift separately to obtain the functions $dn/dz$ and $dn/d\mathcal{M}$. We are particularly interested here in testing different SMBH-host galaxy relations. We therefore construct the function $dn/dz$ and $dn/d\mathcal{M}$ under four different assumptions: (i) model KH13 is constructed by considering both the $M - \sigma$ and $M - M_*$ relations from [14]; (ii) model G09 is based on the $M - \sigma$ relation of [15]; (iii) model S16 employs both the $M - M_*$ and $M - \sigma$ relation from ref. [16]; (iv) model ALL is the combination of all 14 SMBH mass–host galaxy relations listed in Supplementary Table 3. For each of these four models, the allowed regions of $dn/dz$ and $dn/d\mathcal{M}$ are shown in Fig. 3. The figure highlights the large uncertainty in the determination of the SMBHB merger rate and unveils the trend of the chosen models; S16 and KH13 represent the lower and upper bound to the rate, whereas G09 sits in the middle and is representative of the median value of model 'ALL'. These prior bands need then to be described analytically using the parameters of Eq. (2). The shape of these priors and how they differ (or not) from model to model are shown by Supplementary Fig. 3.

We then ensured that once the bands of Fig. 3 are imposed on our model parameters ($\theta = \{\dot{n}_0, \beta, z_*, \alpha, \mathcal{M}_*, e_t(,\eta)\}$), that the resulting distribution of characteristic amplitudes $h_c$ is consistent with that of the original models. We computed the GW background under the assumption of circular GW driven systems (i.e., $h_c \propto f^{-2/3}$) and compared the distributions of $h_{1yr}$, i.e., the strain amplitudes at $f = 1$ yr$^{-1}$. The $h_{1yr}$ distributions obtained with the two techniques were found to follow each other quite closely with a difference of median values and 90% confidence regions smaller than 0.1dex. We conclude that our analytical models provide an adequate description of the observationally inferred SMBHB merger rate and can therefore be used to constrain the properties of the cosmic

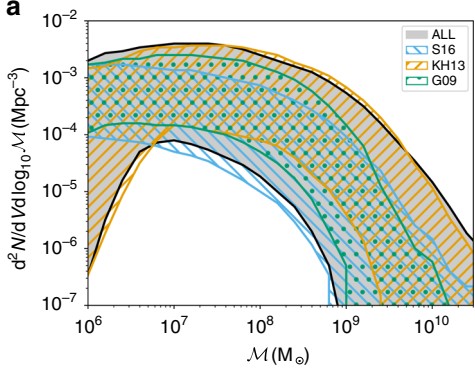
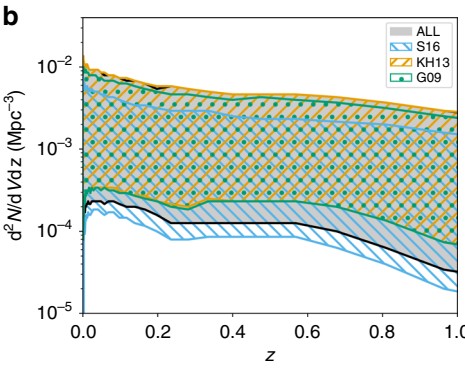

**Fig. 3** Astrophysical prior on the SMBHB chirp mass and redshift distributions. Left panel: **a** mass density distribution $dn/d\mathcal{M}$ of the four astrophysical priors selected in this study (see text for full description). Right panel: **b** redshift evolution of the SMBHB mass density for the same four models. It is noteworthy that the coloured region represent the 99% interval allowed by each model, this is why individual models can extend beyond the region associated to model ALL (which includes KH13, G09, and S16 as subsets)

SMBHB population. In particular model KH13 provides an optimistic prediction of the GW background with median amplitude at $f = 1 \, yr^{-1}$ of $h_{1yr} \approx 1.5 \times 10^{-15}$; model G09 results in a more conservative prediction $h_{1yr} \approx 7 \times 10^{-16}$; model S16 result in an ultra conservative estimate with median $h_{1yr} \approx 4 \times 10^{-16}$; and finally the characteristic amplitude predicted by the compilation of all models (ALL) encompasses almost two orders of magnitudes with median value $h_{1yr} \approx 8 \times 10^{-16}$.

As for the parameters defining the binary dynamics, we assume that all binaries have the same eccentricity for which we pick a flat prior in the range $10^{-6} < e_i < 0.999$ (see Supplementary Fig. 3). In the extended model, featuring a rescaling of the density $\rho_i$ regulating the binary hardening in the stellar phase, we assume a log flat prior for the multiplicative factor $\eta$ in the range $0.01 < \eta < 100$. For more detailed results of including this additional density parameter see Supplementary Table 2, Supplementary Note 1 and Supplementary Fig. 4.

**Likelihood function and hierarchical modelling**. By making use of Bayes theorem, the posterior probability distribution $p(\theta \mid d, H)$ of the model parameters $\theta$ inferred by the data $d$ given our model $H$ is

$$p(\theta|d, H) = \frac{p(d|\theta,H)p(\theta|H)}{\mathcal{Z}_H}, \tag{4}$$

where $p(\theta \mid H)$ is the prior knowledge of the model parameters, $p(d \mid \theta, H)$ is the likelihood of the data $d$ given the parameters $\theta$ and $\mathcal{Z}_H$ is the evidence of model $H$, computed as

$$\mathcal{Z}_H = \int p(d|\theta,H)p(\theta|H)d\theta. \tag{5}$$

The evidence is the integral of the likelihood function over the multi-dimensional space defined by the model parameters $\theta$, weighted by the multivariate prior probability distribution of the parameters. When comparing two competitive models A and B, the odds ratio is computed as

$$\mathcal{O}_{A,B} = \frac{\mathcal{Z}_A}{\mathcal{Z}_B}\frac{P_A}{P_B} = \mathcal{B}_{A,B}\frac{P_A}{P_B}, \tag{6}$$

where $\mathcal{B}_{A,B} = \mathcal{Z}_A/\mathcal{Z}_B$ is the Bayes factor and $P_H$ is the prior probability assigned to model $H$. When comparing the four models KH13, G09, S16 and ALL, we assign equal prior probability to each model. Therefore, in each model pair comparison, the odds ratio reduces to the Bayes factor. Above we have defined the distribution of prior parameters $p(\theta \mid H)$, to proceed with model comparison and parameter estimation we need to define the likelihood function $p(d \mid \theta, H)$.

The likelihood function, $p(d \mid \theta, H)$, is defined following ref. [53]. We take the posterior samples from the Parkes PTA analysis (courtesy of Shannon and collaborators) used to place the 95% upper limit at $h_{1yr} = 1 \times 10^{-15}$, when a single power law background $h_c \propto f^{-2/3}$ is assumed. However, for our analysis we would like to convert this upper limit at $f = 1 \, yr^{-1}$ to a frequency dependent upper limit on the spectrum as shown by the orange curve in Fig. 1. Our likelihood is constructed by multiplying all bins together, therefore the resulting overall limit from these bin-by-bin upper limits must be consistent with $h_{1yr} = 1 \times 10^{-15}$. The $f_{1yr}$ posterior distribution is well fitted by a Fermi function. To estimate a frequency dependent upper limit, we use Fermi function likelihoods at each frequency bin, which are then shifted and re-normalized in order to provide the correct overall upper limit. In our analysis we consider the contributions by only the first four frequency bins of size $1/11 \, yr^{-1}$, as the higher frequency portion of the spectrum provides no additional constraint. We have verified that when we include additional bins the results of the analysis are unchanged. Ideally, we would take the bin-by-bin upper limits directly from the pulsar timing analysis to take account of the true shape of the posterior; however, the method we use here provides a consistent estimate for our analysis.

Having defined the population of merging binaries, the astrophysical prior and the likelihood based on the PPTA upper limit result, we use a nested sampling algorithm[54,55] to construct posterior distributions for each of the six model parameters. For the results shown here, we use 2,000 live points and run each analysis 5 times, giving an average of around 18,000 posterior samples.

**Data availability**. The posteriors are avaliable from www.sr.bham.ac.uk/pta/publications/ncomm2018/posteriors. The code used for the analysis in this study are available from the corresponding author on request.

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

## Acknowledgements

H.M. and A.V. acknowledge the support by the Science and Technology Facilities Council (STFC). S.C. acknowledges the support of the University of Birmingham via the AE Hills scholarship. A.S. is supported by a URF of the Royal Society.

## Author contributions

All the authors have contributed to this work.

## Additional information

**Competing interests:** The authors declare no competing financial interests.

