## [Peer Review File · Nature Communications]

Reviewer #1 (Remarks to the Author):

Middleton et al have moved the manuscript, "No Tension Between Assembly Models of Supermassive Black Hole Binaries and Pulsar Observations" to Nature Communications. Supposedly the authors have responded to the initial reviews provided to them by Nature, but I have read through the current version of the manuscript and found only small, cosmetic changes relative to the original draft, although they have improved the figure presentation. My major comment on ephemerides still stands. Below I include my original review, edited to remove the comments about suitability for Nature and about the figures. Unless the ephemeris issue is at least mentioned, I would not be in favour of publishing this article.

Nature review:

This is a very clear and well-written paper which approaches the question of the implications of pulsar timing array limits on models of the expected numbers of SMBHBs by using Bayesian inference to constrain the parameters of a set of models which correspond to several recently published SMBHB models. By using a single parametrization, the authors facilitate comparisons between the models, which is a very useful contribution to the field.

The context for the paper is the publication of a paper by Shannon et al in 2015 which claimed that a PTA limit derived from 4 pulsars was incompatible with the basic power-law GW signal predicted by the simplest SMBHB models. Various papers since then have discussed the legitimacy of that particular claim. Notably, the NANOGrav 9-year isotropic limit paper (Arzoumanian et al 2016) made the point that the simplest power-law models were already considered somewhat unrealistic, since they completely neglected the likely possibilities of interactions with stars and gas, and binaries with non-zero eccentricities. Middleton et al have now done a good job of quantifying that statement in relation to the Parkes limit. Therefore it is a timely paper and their formalism may see widespread use in future. I would not go so far as to say that it is a very surprising or unexpected result.

A point of substance, which I know the authors will also hear from other sources, is the validity of the Parkes limit in the first place. The authors are certainly aware of the discussions within the recent International Pulsar Timing Array meetings of the use of different Solar System ephemerides and their effect on PTA limits. The Parkes limit used the DE421 ephemeris, as did the recent NANOGrav and EPTA data releases and limits, but this ephemeris has been superseded, with the most recent version being DE436. The authors will know that changing the ephemeris often results in a change in the limit on the gravitational-wave background. While there are not yet any published limits using DE436, I do think the authors should acknowledge the possibility that the Parkes limit might have been influenced by the choice of an ephemeris now known to be less good than others. All that said, the discussion on the astrophysical implications of future, more stringent limits (which would presumably be derived with better ephemerides) is very useful.

Reviewer #2 (Remarks to the Author):

What are the major claims of the paper?

I refereed this paper when it was submitted to Nature (referee #2), so this is my second report. The major claims won't be repeated here, and this report will instead will focus on the changes

that have been made to the paper, in particular with respect to the first referee report.

Are they novel and will they be of interest to others in the community and the wider field?

There are still concerns about the novelty of the results. The authors argue that their results are more robust as they use a more rigorous approach, which is generally true. However, the conclusions have been widely reported elsewhere (including using the same language “no tension” by the same authors.).

A number of suggestions were made about how to broaden the appeal of the article to make of interest to a wider audience that might not be familiar with pulsar timing array research. This includes a better link to the bigger picture and more context.

For example, the paper as written doesn't fully motivate the question “why” supermassive black hole binaries are interesting or important astrophysical objects. For example, the paper doesn't mention how these binaries (and supermassive black holes more generally) fit into the larger picture of our galaxy formation and evolution. This question of “why” needs to be motivated to ensure the paper has a general audience, and is usually done so in the “first paragraph” or first paragraph of the main text. Similarly, the paper doesn't explain explicitly why predictions for the GW amplitude are so uncertain / why these predictions are important.

The paper would also benefit from connection to other wavelengths or to future facilities. The authors chose not to motivate how future measurements of M-sigma or galaxy merger rates could resolve the current wide spread in estimates of these parameters, and could narrow predictions to the point that PTA limits would be useful in resolving discrepancies.

Pulsar timing arrays are one of the cornerstone activities for the 700-million euro Square Kilometre Array (SKA) phase-1 project. I would have expected that if these findings were significant they would have implications for PTA activities on SKA. (Particularly when considering the predictions for a weak GWB that are implicit in this paper).

If the conclusions are not original, it would be helpful if you could provide relevant references. Is the work convincing, and if not, what further evidence would be required to strengthen the conclusions?

In the first report, a number of references which come to similar conclusions as this paper were listed. While the other papers may not have provided as the joint assessment as that in this manuscript, the authors need to acknowledge these interpretations in the main text of the manuscript.

On a more subjective note, do you feel that the paper will influence thinking in the field? Please feel free to raise any further questions and concerns about the paper.

Given other papers have come to similar conclusions, it is unclear what the impact of this paper will be. For example it is likely that Sesana et al. (2016, MNRAS, 463, L6) and Arzoumanian et al. (2016, ApJ, 821, 13) will be referenced regularly when the “no tension” argument is made.

We would also be grateful if you could comment on the appropriateness and validity of any statistical analysis, as well the ability of a researcher to reproduce the work, given the level of detail provided.

Given the lack of access to raw PPTA data sets, the authors have done the best job that they could. They should note in the main text that this is a key assumption of their work.

It is unclear what weighting the authors give to their “comprehensive suite” of models listed in

Table 2: are they all given equal weighting. If so, the authors should state and justify this. The authors state in their response that a "critical assessment of the models goes beyond the scope of the paper". Surely a critical assessment was made when including the models in the first case? Have the authors rejected other predictions for merger rates and M-sigma relations, if so, why?

There is still concern about the turnover at low frequency. While the authors note in the main text (line 99) that a power-law spectrum is consistent with their bands in figure 1, the departure from power law is significant and would likely significantly change Bayes factors, K-L divergences, etc. Indeed, Shannon et al. pointed out that a turnover could be relieved by having a low-frequency turnover.

Other comments:

Line 12: upperlimit -> upper limit

Line 19: This sentence is very technical and will be hard for a general audience to interpret.

Line 25: First paragraphs usually end in a general statement about the implications of the findings that can be understood by a general audience.

Line 102: Figures are mentioned out of order.

Line 112: Least favourite -> Least favoured

Line 135: Taylor et al. (2016, ApJL 819, 6) also make argue against the high cadence observing strategy. This needs to be referenced.

Line 156: Please define quantitatively what "very low" means in terms of frequency.

We thank the Referees for their constructive comments and remarks that helped improving the quality of the paper. We revised the manuscript accordingly trying to address all points as thoroughly as we could. We hope the Referees and Editor are satisfied with the current version. Answers to each individual point are presented below.

Reviewer #1 (Remarks to the Author):

Middleton et al have moved the manuscript, "No Tension Between Assembly Models of Supermassive Black Hole Binaries and Pulsar Observations" to Nature Communications.

Supposedly the authors have responded to the initial reviews provided to them by Nature, but I have read through the current version of the manuscript and found only small, cosmetic changes relative to the original draft, although they have improved the figure presentation. My major comment on ephemerides still stands. Below I include my original review, edited to remove the comments about suitability for Nature and about the figures. Unless the ephemeris issue is at least mentioned, I would not be in favour of publishing this article.

Nature review:

This is a very clear and well-written paper which approaches the question of the implications of pulsar timing array limits on models of the expected numbers of SMBHBs by using Bayesian inference to constrain the parameters of a set of models which correspond to several recently published SMBHB models. By using a single parametrization, the authors facilitate comparisons between the models, which is a very useful contribution to the field.

The context for the paper is the publication of a paper by Shannon et al in 2015 which claimed that a PTA limit derived from 4 pulsars was incompatible with the basic power-law GW signal predicted by the simplest SMBHB models. Various papers since then have discussed the legitimacy of that particular claim. Notably, the NANOGrav 9-year isotropic limit paper (Arzoumanian et al 2016) made the point that the simplest power-law models were already considered somewhat unrealistic, since they completely neglected the likely possibilities of interactions with

stars and
gas, and binaries with non-zero eccentricities. Middleton et al have now
done a
good job of quantifying that statement in relation to the Parkes limit.
Therefore
it is a timely paper and their formalism may see widespread use in future.
I would
not go so far as to say that it is a very surprising or unexpected result.

A point of substance, which I know the authors will also hear from other
sources, is
the validity of the Parkes limit in the first place. The authors are
certainly
aware of the discussions within the recent International Pulsar Timing
Array
meetings of the use of different Solar System ephemerides and their effect
on PTA
limits. The Parkes limit used the DE421 ephemeris, as did the recent
NANOGrav and
EPTA data releases and limits, but this ephemeris has been superseded, with
the most
recent version being DE436. The authors will know that changing the
ephemeris often
results in a change in the limit on the gravitational-wave background.
While there
are not yet any published limits using DE436, I do think the authors should
acknowledge the possibility that the Parkes limit might have been
influenced by the
choice of an ephemeris now known to be less good than others. All that
said, the
discussion on the astrophysical implications of future, more stringent
limits (which
would presumably be derived with better ephemerides) is very useful.

Answer:

We felt reluctant to quote a result which is not yet published, however we
have added
the following text and provided a reference to Hobbs & Dai 2017, which
briefly mentions
the result without a quantitative statement.
"It has been noted, however, that these limits start to be sensitive to
uncertainties in the determination of the solar system ephemeris used in
the analysis. Recent unpublished work has in fact found that different
ephemeris choices can result in a partial degradation of the upper limit
~\cite{HobbsDai:2017}. This is still an active area of research which may
lead to a small upward revision of the upper limit, a circumstance which,
if anything, will strengthen the conclusion of our analysis."

Reviewer #2 (Remarks to the Author):

I refereed this paper when it was submitted to Nature (referee #2), so this
is my
second report. The major claims won't be repeated here, and this report
will

instead will focus on the changes that have been made to the paper, in particular with respect to the first referee report.

Question 1:

There are still concerns about the novelty of the results. The authors argue that their results are more robust as they use a more rigorous approach, which is generally true. However, the conclusions have been widely reported elsewhere (including using the same language "no tension" by the same authors.).

Answer:

We stress once again that although similar conclusions have been already reported, this is the first systematic study taking into account all relevant physics shaping the GW signal and tries to constrain it via a fully Bayesian hierarchical inference. As the referee also recognizes our approach is more robust, general and rigorous than previous analysis. For comparison Arzoumanian et al. 2016, only consider one physical ingredient at a time, and uses a dataset that provides a 50% higher upper limit at $1.5E-15$. For example, the MBH-galaxy relation is constrained by assuming a single power law model, which is also the case in Simon & Burke Spolaor 2016. Finally, Sesana et al. 2016 is a theoretical paper that does not present any analysis of the data. The paper only states that a putative selection bias that causes a lower M-sigma relation, would result in a lower GWB that would completely remove any tension with PTA upper limits. Here, by considering a comprehensive ensemble of models, we rigorously show that a selection bias in the MBH mass estimates is not required (nor is a turnover due to eccentricity or environment) to reconcile PTA non detection and SMBHB populations.

We therefore believe that there is no comparably rigorous analysis in the literature and certainly not with the required resonance to counter-balance some bald claims appeared in Shannon et al. 2015. This work should become the reference standard for the state of the art of astrophysical inference from PTAs and its publication in the prestigious Nature Communications will guarantee an appropriate visibility within the physics and astronomy community.

Question 2:

A number of suggestions were made about how to broaden the appeal of the article to make of interest to a wider audience that might not be familiar with pulsar timing array research. This includes a better link to the bigger picture and more context. For example, the paper as written doesn't fully motivate the question "why" supermassive black hole binaries are interesting or important astrophysical objects. For example, the paper doesn't mention how these binaries (and supermassive black holes more generally) fit into the larger picture of our galaxy formation and evolution. This question of "why" needs to be motivated to ensure the paper has a general audience, and is usually done so in the "first paragraph" or first paragraph of the main text.

Similarly, the paper doesn't explain explicitly why predictions for the GW amplitude are so uncertain / why these predictions are important.

Answer:

This is an important point since a proper framing of the result in the general context of galaxy formation is indeed missing. Taking advantage of the extra space allowed by Nature Communication, we now open the introduction with the following paragraph that describes the cosmic role played by SMBHBs and the importance of detecting their GWs:
"Dedicated timing campaigns of ultra-stable radio pulsars lasting over a decade and carried out with the best radio telescopes around the globe have targeted the isotropic gravitational-wave (GW) background in the frequency region $\hat{\sim} 10^9 \hat{\sim} 10^7$ Hz generated by the cosmic population of merging super massive black hole binaries (SMBHBs). In the hierarchical clustering scenario of galaxy formation, galaxies form through a sequence of mergers (White & Rees 1978). In this process, the SMBHs hosted at their center will inevitably form a large number of binaries (Begelman et al. 1980), forming an abundant population of GW sources in the Universe. Detecting and/or placing constraints on their emitted signal will thus provide an insight into the formation and evolution of SMBHBs in connection with their galaxy hosts and will help to better understand the role played by SMBHBs in galaxy evolution and the dynamical processes operating during galaxy mergers (see Sesana 2013a, for a review)."

Question 3:

The paper would also benefit from connection to other wavelengths or to future facilities. The authors chose not to motivate how future measurements of M - σ or galaxy merger rates could resolve the current wide spread in estimates of these parameters, and could narrow predictions to the point that PTA limits would be useful in resolving discrepancies.

Answer:

Investigating how observations at different wavelengths might narrow down the allowed range of the GWB level would be a quite extensive work, worth of a separate project. Certainly, as the SMBH-galaxy relation or the galaxy merger rate become more constrained, the expected range of the signal will narrow down, and possibly other effects (e.g. dynamics, merger timescales, etc) will be the dominant source of uncertainty. A stringent upper limit or a detection will therefore allow to constrain those parameters. Since we concentrate on different SMBH-galaxy relations here, we added as an example a brief discussion on how more stringent determination of this relation with ELT or TMT will allow to better constrain the galaxy merger rate. We

added the following paragraph to the discussion, along with the relevant references:

"Future facilities such as the Extremely Large Telescope (Gilmozzi & Spyromilio 2007) and the Thirty Meter Telescope (Sanders 2013) will likely measure many more SMBH masses in elliptical galaxies (Do et al. 2014), providing a better understanding of the SMBH-host galaxy relations. PTA limits may therefore be used to gain more information about the other underlying uncertainties in the model, in particular the massive galaxy merger rate, which is currently poorly constrained observationally (see, e.g., Lotz et al. 2011; Mundy et al. 2017)."

Question 4:

Pulsar timing arrays are one of the cornerstone activities for the 700-million euro Square Kilometre Array (SKA) phase-1 project. I would have expected that if these findings were significant they would have implications for PTA activities on SKA. (Particularly when considering the predictions for a weak GWB that are implicit in this paper).

Answer:

We argue that our findings are significant on their own regardless of future implications for, e.g., SKA. We think it is important to highlight that correct inference from current data does not warrant major issues with our understanding of SMBHs, as has been claimed. Nonetheless, it is true that our results mildly favour (quite obviously, since we are making inference from an upper limit) lighter SMBHs for fixed galaxy properties. This means that the GWB might be quite below the 10^{-15} value, which indeed highlights the importance of going beyond current facilities and adding more good timers to the PTAs. We highlighted this by adding the following paragraph in the discussion, along with the relevant references:

"This indicates that the gravitational wave background level is likely below the 10^{-15} level, making detection difficult with current telescopes. In this respect, our analysis highlights the importance of upcoming facilities such as MeerKAT (Booth et al. 2009), FAST (Nan et al. 2011) and the Square Kilometer Array (SKA Dewdney et al. 2009). Their superior timing capabilities, together with their survey potential in finding new stable millisecond pulsars, will provide the necessary ground to improve sensitivity down to $h \sim 10^{-16}$, which is in line with the lower limit of the expected stochastic gravitational wave background according to our current understanding of SMBH evolution (Bonetti et al. 2017)."

Question 5:

In the first report, a number of references which come to similar conclusions as this paper were listed. While the other papers may not have provided as the joint assessment as that in this manuscript, the authors need to acknowledge these interpretations in the main text of the manuscript.

Answer:

We acknowledged previous analysis and compared them to ours in the discussion session, highlighting in what respect our analysis is different and therefore more robust. We added the following paragraph in Section 3: "Compared to previous analyses, our work implies a stronger rejection of the statement that there is tension between PTA data and theoretical SMBHB population models. For example Simon & Burke-Spolaor (2016) invoked time delays to reconcile the PPTA upper limit with selected SMBH-galaxy relations, however they assume a narrow range of possible SMBHB merger histories and do not consider SMBHB dynamics. The Arzoumanian et al. (2016) analysis tends to favour a spectral turnover due to either high eccentricity or strong environmental coupling, however they use a simplified analysis where each relevant physical parameter is accounted for separately. When allowing all the parameters to vary simultaneously, we find that none of them has a critical impact on the inference, and current SMBHB population models are broadly consistent with the PTA upper limits, without the need to invoke a low frequency spectral turnover."

Question 6:

Given other papers have come to similar conclusions, it is unclear what the impact of this paper will be. For example it is likely that Sesana et al. (2016, MNRAS, 463, L6) and Arzoumanian et al. (2016, ApJ, 821, 13) will be referenced regularly when the "no tension" argument is made.

Answer:

We believe that our answer to Question 1 also addresses this point. The analysis presented here is more rigorous, robust and general than any other attempt of astrophysical inference from PTA data and will set the bar for the years to come. We'd like to stress once again that Sesana et al. 2016 is a theoretical paper that investigates a specific set of theoretical models that remove any putative tension with PTA observations and that Arzoumanian et al. performs a much simpler analysis, isolating individual physical ingredients (thus ignoring correlations among them) and using a

limit that is not as stringent as the PPTA one. Moreover, several people still quote the original Shannon et al. 2015 mentioning a tension between PTA data and models. We believe that publishing a state of the art analysis correcting some of those claims in a visible journal like Nature Communication will have a strong positive impact to the wide community.

Question 7:

Given the lack of access to raw PPTA data sets, the authors have done the best job that they could. They should note in the main text that this is a key assumption of their work.

Answer:

In line with the Nature Communications style, we have made a new section in the main text for methods. Much of the contents of appendix A has been moved to this section of the main text, including our description of how we construct the likelihood and that a more ideal scenario would be to directly use the bin-by-bin PPTA upper limit to take full account of the shape of the true posterior. This reads as:

"The likelihood function, $p(d|\hat{f}, M)$ is defined following Chen et al. (2017a). We

take the posterior samples from the Parkes PTA analysis (courtesy of Shannon and collaborators) used to place the 95% upper limit at $h_{1\text{yr}} = 1 \times 10^{-15}$, when a single

power law background $h_c \hat{f}^{2/3}$ is assumed. However, for our analysis we would

like to convert this upper limit at $f = 1\text{yr}^{-1}$ to a frequency dependant upper limit on

the spectrum as shown by the orange curve in figure 1. Our likelihood is constructed

by multiplying all bins together, therefore the resulting overall limit from these bin-

by-bin upper-limits must be consistent with $h_{1\text{yr}} = 1 \times 10^{-15}$. The $f_{1\text{yr}}$ posterior

distribution is well fitted by a Fermi function. To estimate a frequency dependant

upper limit, we use Fermi function likelihoods at each frequency bin, which are then

shifted and re-normalised in order to provide the correct overall upper limit. In

our analysis we consider the contributions by only the first 4 frequency bins of size

$1/11\text{ yr}^{-1}$, as the higher frequency portion of the spectrum provides no additional

constraint. We have verified that when we include additional bins the results of the

analysis are unchanged. Ideally, we would take the bin-by-bin upper limits directly

from the pulsar timing analysis to take account of the true shape of the posterior;

however, the method we use here provides a consistent estimate for our analysis."

Question 8:

It is unclear what weighting the authors give to their "comprehensive suite" of models listed in Table 2: are they all given equal weighting. If so, the authors should state and justify this. The authors state in their response that a "critical assessment of the models goes beyond the scope of the paper". Surely a critical assessment was made when including the models in the first case? Have the authors rejected other predictions for merger rates and M-sigma relations, if so, why?

Answer:

We expanded the description of the models in the Method section, giving more details about the ensemble of models considered and the various assumption we make. The full amended text is:
"For each of the galaxy mass functions and pair fractions we consider three estimates given by the best fit and the two boundaries of the 1 σ confidence interval reported by the authors. We therefore have 12 \times 12 \times 2 = 288 galaxy merger rates. Each merging galaxy pair is assigned SMBHs with masses drawn from 14 different SMBH-galaxy relations found in the literature, for more details see Supplementary Table A.3). SMBHBs are assumed to merge in coincidence with the host galaxies (i.e. no stalling or extra delays), but can accrete either before or after merger according to the three different prescriptions described in Sesana et al. (2009). This gives a total of 14 \times 3 = 42 distinctive SMBH population for a given galaxy merger model. We combine the 288 galaxy merger rates as per equation (3) and the 42 SMBH masses assigned via using Supplementary Table A.3, plus accretion prescriptions into a grand total of 12096 SMBHB population models. Given the uncertainties, biases, selection effect, and poor understanding on the underlying physics affecting each of the individual ingredients, we do not attempt a ranking of the models but we give each of them equal weight. The models result in an allowed SMBHB merger rate density as a function of chirp mass and redshift."
The referee can appreciate that the decision of giving equal weight to all models is necessarily imposed by the impossibility of assessing the uncertainties that go in the different assumptions. For example the 14 MBH-galaxy relations are divided in MBH-sigma and MBH-bulge mass relations. Whether one is more fundamental than the other is a matter of debate and an unsettled issue. Within the single subsets, relations have been derived using different samples, and measurement techniques. For example σ can be measured with different apertures (half light radius, effective radius, fixed angular aperture, etc), causing possible biases. Trying to quantifying whether one relation is better than another is probably not even possible within our current knowledge.
We tried to make a comprehensive list of the most popular relations found in the literature, covering all possibilities. We did not intentionally

exclude any model that could significantly change our results. We also notice that the GW signal of our suit of models spans a range that covers the predictions of other models constructed in different ways. We stressed this in the introduction by adding the sentence:

"Moreover, Model \hat{h} spans an h 1yr amplitude range that comfortably include

GW backgrounds estimated by other authors employing different techniques (e.g. McWilliams et al. 2014; Ravi et al. 2015; Kulier et al. 2015; Kelley et al. 2017)."

along with the relevant references.

Finally, "We highlight the impact of the SMBH-galaxy relation by considering a selection of models which cover the entire range of the predicted background amplitude." as specified in the introduction.

Question 9:

There is still concern about the turnover at low frequency. While they authors

note in the main text (line 99) that a power-law spectrum is consistent with their

bands in figure 1, the departure from power law is significant and would likely

significantly change Bayes factors, K-L divergences, etc. Indeed, Shannon et al.

pointed out that a turnover could be relieved by having a low-frequency turnover.

Answer:

Although the spectrum significantly departs from a single power law, this occurs mostly at high frequencies. Except for a small bending close to 1nHz (where in any case current PTAs have no sensitivity), the spectra show a prominent steepening at $f > 10\text{nHz}$, due to small number statistics of the most massive sources generating the GWB, as described in Sesana, Vecchio & Colacino 2008. In fact, in the region of PPTA maximum sensitivity, the spectra follow the standard $f^{-2/3}$ power law. Having said that, all the Bayes factors and K-L divergences are computed considering the PPTA upper limit and power of the signal at each individual frequency and bin, and therefore consistently include any information about the spectral shape of the signal. It is certainly true that a low frequency turnover can relieve any tension with the data, but one of the main results of our study is that this is not required. The upper limit is consistent with signals with or without turnover alike.

In the results section, we added this extra description of the actual shape of the bands shown in figure 1:

"The confidence bands across the frequency range are constructed by taking the relevant

credibility region of the posterior distribution of h_c at each frequency, and therefore the

boundaries of each band do not follow any particular functional form as a function of frequency."

Other comments:

Question:

Line 12: upperlimit -> upper limit

Answer: Corrected

Question:

Line 19: This sentence is very technical and will be hard for a general audience to interpret.

Answer:

We have removed this sentence from the abstract.

Question:

Line 25: First paragraphs usually end in a general statement about the implications of the findings that can be understood by a general audience.

Answer:

We have added an additional sentences to motivate overall study:
"Here we consider the most stringent upper limit from the PPTA in order to glean what can be learnt at this stage and also determine whether current SMBHB population models are indeed cast into doubt. We find that although PTA's have well and truly entered the region where detection is considered possible from model predictions, as yet, there no reason to scrap these models."

Question:

Line 102: Figures are mentioned out of order.

Answer: We have rearranged the text in this section to improve clarity and ordering for referencing figures.

Question:

Line 112: Least favourite -> Least favoured

Answer: Corrected

Question:

Line 135: Taylor et al. (2016, ApJL 819, 6) also make argue against the high cadence observing strategy. This needs to be referenced.

Answer: We now referenced Taylor et al. 2016 in the discussion

Question:

Line 156: Please define quantitatively what "very low" means in terms of frequency.

Answer: that was a generic reference to the low frequency PTA band. We rephrased the sentence as "which in turn leads to a legitimate re-thinking

of the PTA observing strategy to target possibly more promising frequencies of the GW spectrum."

Reviewer #1 (Remarks to the Author):

I have read the third version of the manuscript, "No Tension Between Assembly Models of Supermassive Black Hole Binaries and Pulsar Observations" by Middleton et al. I am satisfied with the authors' responses to my previous comments and have no major complaints with the new, mostly clarifying material that they have added. I therefore think this paper is now suitable for publication in Nature Communications.